# Geochemistry and Acid Hydrometallurgy Accessibility of Uraninite from Mianhuakeng Granite-Hosted Uranium Deposit, South China

**Jian Wang** [1,2] , **Zhanxue Sun** [2,*], **Guangrong Li** [2,*], **Yajie Liu** [2], **Zhongkui Zhou** [2],
**Xuegang Wang** [2], **Zhihong Zheng** [2], **Yipeng Zhou** [2], **Kai Zhao** [2], **Ling Xiang** [2] and **Jiaxin Wei** [2]

[1]  School of Water Resources and Environment, Beijing Key Laboratory of Water Resources and Environmental
   Engineering, and MOE Key Laboratory of Groundwater Circulation and Environmental Evolution, China
   University of Geosciences (Beijing), Beijing 100083, China; wangjiangoodman@163.com

[2]  State Key Laboratory of Nuclear Resources and Environment, East China University of Technology,
   Nanchang 330013, China; yjliu@ecut.edu.cn (Y.L.); zhkzhou@ecut.edu.cn (Z.Z.); xgwang@ecut.edu.cn (X.W.);
   zhzheng@ecut.edu.cn (Z.Z.); zhouyp@ecut.edu.cn (Y.Z.); 18770915836@163.com (K.Z.);
   lingxiang326@126.com (L.X.); juliana0204_1994@163.com (J.W.)

*  Correspondence: zhxsun@ecut.edu.cn (Z.S.); liguangrong0086@ecut.edu.cn (G.L.);
   Tel.: +86-791-83897597 (Z.S.); +86-791-83898197 (G.L.)

**Abstract:** Systematic study of the surface chemical properties of uranium minerals is necessary
to improve the uranium ore extracting process. The presented work aims to argue geochemistry
and acid hydrometallurgy accessibility of uraninite from the Mianhuakeng (MHK) granite-hosted
uranium deposit, South China, which provides insight on this ore extracting domain. Mineralogy,
geochemical composition, U–Th–Pb chemical age, and uranium deportment of the uraninite were
systematically analyzed by using scanning electron microscope with energy dispersion spectrum
(SEM-EDS), an electron probe microanalyzer (EPMA), and x-ray photoelectron spectroscopy (XPS).
The results showed that uraninite was intergrowth with coffinite, probably due to uraninite being
partly metasomatized into coffinite along the fissures. The major element content of uraninite such as
for $UO_2$, $SiO_2$, and $CaO$ were $79.46 \pm 2.03$ wt%, $6.19 \pm 1.36$ wt%, and $5.09 \pm 0.80$ wt%, respectively.
Single-point U–Th–Pb chemical ages for uraninite grains were calculated with the EPMA data, and the
results showed ages ranging from a few million to dozens of million years, indicating Pb loss after
uraninite formed. Uranium deportment in uraninite generally existed in the forms of $UO_2$, $U_3O_8$,
and $UO_3$, and mostly showed high valence states suggested by XPS. Uranium on the surface of the
uraninite grain was partially oxidized by sulfuric acid leaching, which led to tetravalent uranium
converting to hexavalent uranium, suggesting uraninite in the MHK uranium deposit is accessible to
be leached by sulfuric acid.

**Keywords:** uraninite; geochemical composition; uranium valence; acid hydrometallurgy;
Mianhuakeng uranium deposit

---

## 1. Introduction

As one type of mostly potential and practical non-fossil energy sources, nuclear energy has
irreplaceable advantage, not only because of low carbon dioxide emissions, but also in terms of operating
stability relative to traditional fossil energy and renewable energy [1]. Therefore, many countries in the
world, especially in China, have paid more attention to the development of nuclear power in order to
meet the international emissions reduction commitment and construct an ecological society [2]. In
recent years, uranium (U) resource consumption has been substantially increased, accompanied by the

ever-increasing need for nuclear power worldwide [3,4]. However, it is a hard fact that uranium ore grades have rapidly declined over the past few decades [3]. Mineral processing techniques, which are used to extract uranium from low grade refractory uranium ores, have become a hot spot and are now receiving renewed attention [5–10]. However, nowadays, systematic study of the surface chemical properties of uranium minerals is still rare, which seriously slow down the uranium ore extracting process techniques [11]. In particular, it is still not very clear as to how the uranium valence changes during the acid leaching of uranium minerals.

According to the host rocks, the uranium deposits in Southern China can be mainly divided into three types: granite-hosted, volcanic-hosted, and carbonaceous-siliceous-pelitic rock-hosted deposits [12,13], of which the first type has attracted the most scientific attention. Over the past decade, research has focused on the characterization of granite-host uranium deposits for the ore genesis, chemical composition, and ore forming age in Southern China. The ore minerals are mainly uraninite, coffinite, and brannerite [14]. $UO_2$ contents in uraninite compositions usually vary from 77.1 to 90.9 wt%, $SiO_2$ from ca. 0.1 to 7.8 wt%, and relatively high CaO from ca. 2.0 to 11.5 wt% [15,16]. Furthermore, abundant studies have been carried out on the ore forming ages of uranium deposits including in situ high sensitivity and high spatial resolution dating techniques [17–21]. Recently, a robust Secondary Ion Mass Spectrometry (SIMS) U–Pb age of ~2.0 Ma was first discovered in the Menggongjie uranium deposit, which has been interpreted as a newly identified mineralization stage, making the Menggongjie deposit the youngest granite-hosted uranium deposit [16]. The predominant valence states of uranium are U(III), U(IV), U(V), and U(VI) across various geological bodies on Earth, which occur widely in compounds of multiple valences. Among these compounds, U(IV) and U(VI) are relatively common in nature [22]. U(IV) is usually reserved as uraninite ($UO_2$) during magmatic, metamorphic, and hydrothermal processes. U(VI) easily dissolves in water in forms of either uranyl compounds or secondary uranium minerals as sulfate, carbonate, vanadates, and phosphate in environments such as sedimentation, evaporation, and oxidation [23]. Native uranium, which was thought to not exist in nature due to its instability and changeable valence of uranium, was first discovered in hydrothermal Guidong and Zhuguang uranium deposits [24]. Meanwhile, it was found that the main uranium minerals in the Xianshi uranium deposit may not be uraninite, otherwise, it might be the first discovery of vorlanite $(CaU)O_4$ enriched by a deposit of economic size [25,26].

In this work, uraninite from the MHK uranium deposit, one of the main uranium minerals, was taken as the target mineral. The petrography and mineralogy, geochemical composition, chemical U–Th–Pb ages and uranium valence of uraninite collected from the MHK (also called No. 302) uranium deposit were systematically analyzed using SEM-EDS, EPMA, and XPS. In particular, the variation characteristics of the uranium valence are discussed in detail when the pristine uraninite samples were leached by chemically pure sulfuric acid. The goal of this research is to provide fundamental knowledge on the surface chemical properties of uraninite from MHK, and to determine how sulfuric acid will oxidize uranium minerals during acid leaching.

## 2. Materials and Methods

### 2.1. Geological Setting and MHK Uranium Deposit

South China, which is composed of the Yangtze Block in the north and the Cathaysian Block in the southeast, has yielded the most important granite-hosted uranium deposits in China, especially in the area of Motianling, Miaoershan, Jiuyishan, Liuchen, Guidong, Zhuguang, Taoshan, Aigao, and Dafuzu [12–14]. The Zhuguang complex, which is composed of multiperiodic granites, is located in the northern Guangdong Province of South China. The MHK deposit is located in the central part of the southern Zhuguang complex, which is pinched between the Mianhuakeng fault (strike 50°–60° N) and Youdong fault (strike 295°–310° N). Tectonically, this location is in the transition zone of the Cathaysian Caledonian uplift and Guangdong Hercynian–Indosinain depression (Figure 1).

The crustal evolution has experienced four tectonic cycles, namely Caledonian, Hercynian Indosinian, Yanshan, and Xishan tectonic cycles, showing the evolution model of activity–stability–reactivation. The Caledonian movement at the end of the early Paleozoic resulted in tight linear compound folds in the lower Paleozoic strata and widespread regional metamorphism, forming the basic outline of the North-East (NE) trending post Caledonian uplift of Fujian and Jiangxi Provinces [14,27,28]. During the Variscan, the whole area was relatively stable, during which a slow ascending and descending movement occurred, showing that the Longtan Formation of the upper Permian was parallel unconformity on the Dangchong Formation of the lower Permian. During the Indosinian movement between the Middle and Late Triassic, the early Paleozoic strata underwent another transformation, while the late Paleozoic strata produced surface sliding and formed transitional folds. With this movement, the region further uplifted, connecting the Jiufeng and Jiulianshan uplifts, thus forming the basic outline of the post Indosinian landmass in northern Guangdong [29]. The Yanshanian movement is characterized by strong fault block activity and large-scale acid magmatic intrusion. The Himalayan movement took place between the Paleogene, Neogene, and the end of the Neogene, which is the continuation and development of Yanshan movement. With the exception of the Silurian, strata of different ages from the Sinian to Quaternary were well developed, which were distributed in the edge of rock mass. Among them, the Paleozoic is the most widely distributed, followed by the Mesozoic and Cenozoic. There were frequent magmatic activities in the area including Indosinian and Yanshanian granites, which were dominated by late Indosinian medium coarse-grained porphyritic biotite granite and early Yanshanian medium coarse-grained biotite granite, followed by the second stage of Indosinian medium coarse-grained porphyritic monzogranite, the third stage of early Yanshanian fine-grained porphyritic biotite granite, and late Yanshanian period fine grained biotite granite with a small amount of late Yanshanian intermediate-basic dike and fine-grained dike intrusion [28].

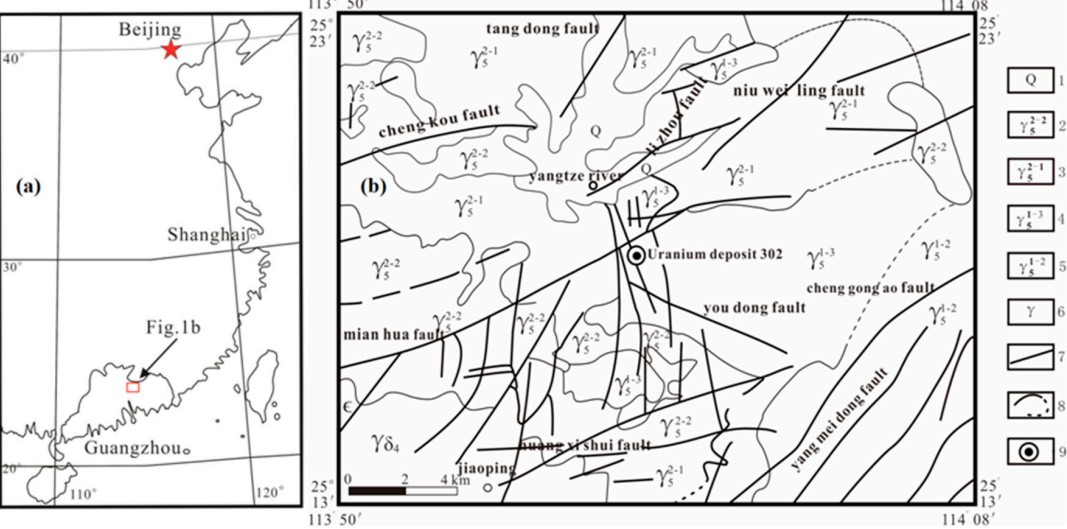

**Figure 1.** (**a**) Schematic map showing the location of the Mianhuakeng uranium deposit. (**b**) Geological sketch map of Mianhuakeng (No. 302) uranium deposit (after Huang et al. [30]). 1—Quartenary; 2—Medium-fine grained biotite-muscovite granite; 3—Medium-grained (porphyritic) biotite granite; 4—Medium-grained (porphymic) biotite-muscovite granite; 5—Medium-coarse grained porphyritic biotite monzonitic granite; 6—Granodiorite; 7—Fault; 8—Measured and inferred geological boundaries; 9—Uranium deposit.

The MHK ore field hosts significant uranium deposits such as those at Shulouqiu, Mianhuakeng, Youdong, Changpai, and Changkeng, etc. (Figure 2), of which the MHK is the largest in South China. There are three groups of faults in the MHK ore field: the Norh-East-East (NEE, probably the ore controlling structure), Norh-West-West (NWW), and Norh-Norh-West (NNW) strike. The predominant

uranium mineral in this ore field is uraninite associated with quartz, fluorite, pyrite, and minor carbonates, which mainly occur in the veins. The uranium mineralization was accompanied by chloritization, hematization, and sericitization, etc.

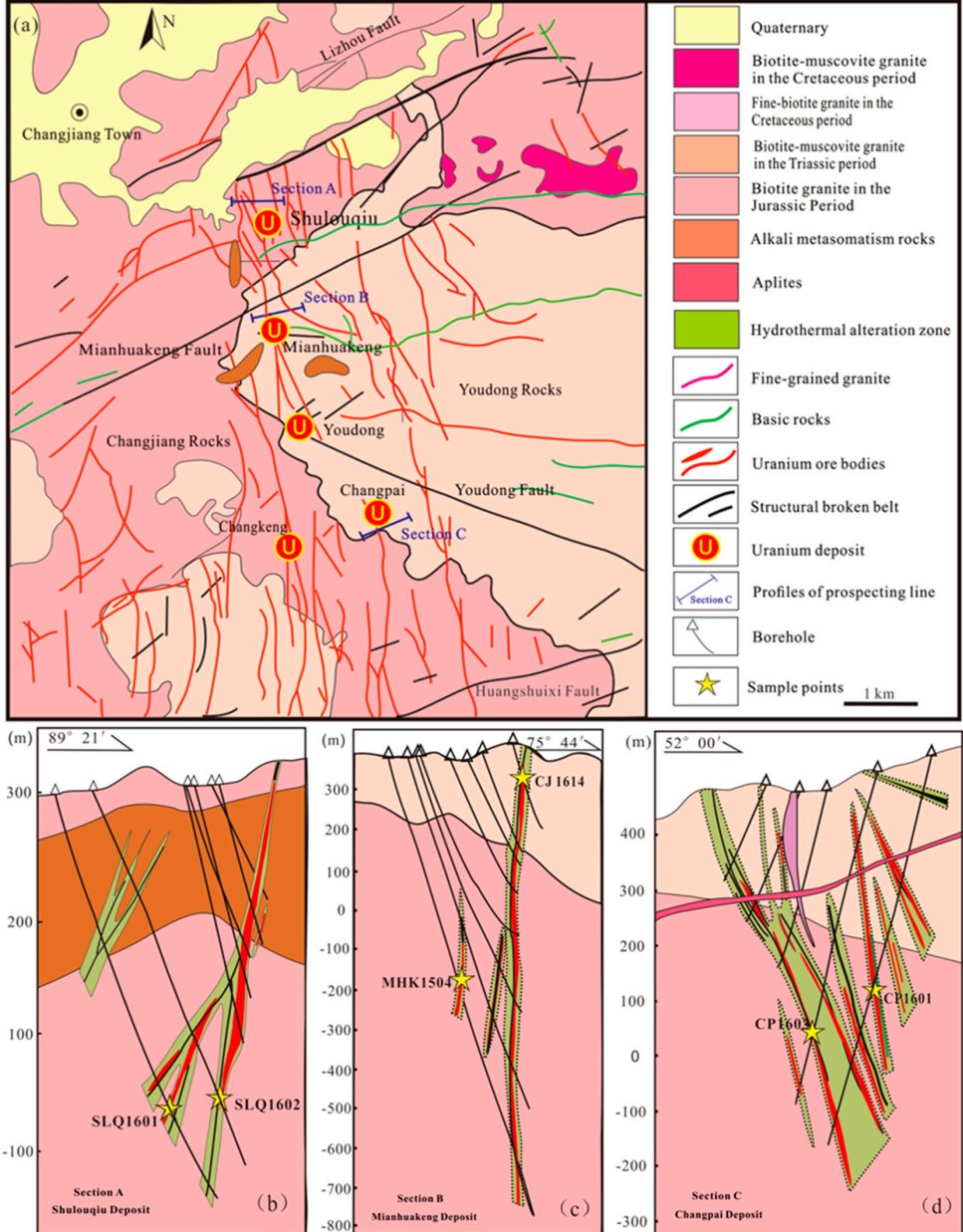

**Figure 2.** The MHK uranium ore field and related geological profiles [27,31]. (**a**) Geological sketch map of MHK uranium ore field. (**b**) Geological cross section of A, Shulouqiu Deposit. (**c**) Geological cross sectionof B, Mianhuakeng Deposit. (**d**) Geological cross sectionof C, Changpai Deposit.

For the MHK uranium deposit precisely, mineralization of the MHK uranium deposit was mainly controlled by the fracture system with a south–north extension, which led to the ore bodies being mainly vein-like and lenticular. These fracture systems probably formed approximately 90 Ma ago, and the major mineralization temperature ranged from 150 to 350 °C, according to fluid inclusion experiments [24]. The buried depth of the ore bodies ranged from the surface to a few hundred meters (Figure 2c), however, the industrial ore body is produced at an elevation from 500 m to −647 m. There are three main deportments of uranium, namely, (1) independent mineral of uranium; (2) isomorphism in accessory minerals; and (3) distribution in other rock forming minerals [32,33]. The ore minerals are

mainly uraninite with a small amount of secondary uranium minerals [13]. The ores are dark red and gray, and uraninite occurs in mostly disseminated veinlet and agglomerate forms (Figure 3). The main gangue minerals are quartz and feldspar (microcline feldspar and plagioclase) and the secondary gangue minerals are sericite, hematite, pyrite, fluorite, calcite, galena, etc. [34]. The main types of alteration are as follows progressively: silicification, hematite mineralization, strong sericitization chloritization, sericitization kaolinization, and weak altered granite [35,36].

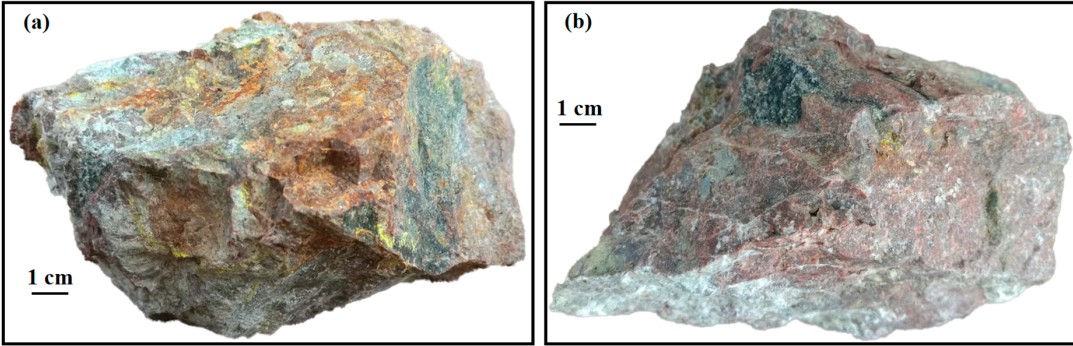

**Figure 3.** Specimen of the uranium ore from MHK. (**a**) The yellow and dark red minerals are oxidized uranium mineral and hematite. (**b**) The green dark minerals are agglomerate uraninite.

Furthermore, wall rock of the upper part of the deposit is medium-grained porphyritic dimicite granite of Youdong Indosinian pluton, with a diagenetic age of $232 \pm 4$ Ma. Moreover, the middle-lower surrounding rock is medium-grained biotite granite of Changjiang Yanshanian plutons, with a diagenetic age from 155 to 164 Ma [31,37,38]. It is noteworthy that uranium minerals are commonly associated with several generations of quartz, carbonate, fluorite, pyrite, and other gangue and alteration minerals. The existence of all these minerals suggests that the hydrothermal ore-forming fluid composition is complicated, and was formed in multiple environments [24]. In this research, samples of U-ore were all collected from the MHK uranium deposit below a depth of 150 m. Then, the samples were mashed and passed through a 50-mesh sieve. Finally, the uraninite was picked out within a binocular eyepiece to make mineral targets that were used to analyze the microscopy, mineral, chemical compositions, and uranium valence.

## 2.2. Analytical Methods

The characterization of the mineral and chemical compositions of the samples was determined at the State Key Laboratory of Nuclear Resources and Environment, East China University of Technology. The microstructure and microzone chemical properties of the uraninite grains were analyzed by the Nova Nano-SEM 450 scanning electron microscope equipped with energy dispersive spectrometry (Inca Energy X-Max20, made by Oxford Instruments, Abingdon, UK). Based on petrographic observations and SEM-EDS analysis, the same uraninite grains were chosen for quantitative chemical analysis and map scanning by a JEOL JXA-8530 electron probe microanalyzer (EPMA). The EPMA analytical conditions were 15 kV accelerating voltage, and a 20 nA beam current with a 5 μm beam diameter for analyzing the uranium oxides composition. Data were reduced using the X-PHI procedure, and detection limits for these elements were approximately 0.01 wt%. Moreover, prior to SEM-EDS and EPMA analysis, thin sections of samples were gold and carbon coated to create a conductive surface, respectively.

XPS was chosen to characterize the uranium chemical valences of uraninite and the relative proportions of different valence uranium in the uraninite. In order to investigate how sulfuric acid oxidized uranium mineral, the target minerals were leached by $3$ g $L^{-1}$ chemically pure sulfuric acid (50 μL) for 2 h before it was tested again. Furthermore, the XPS analysis could also provide additional elemental information on the surface of the uraninite. In this study, the XPS analysis was carried out at

the Guangzhou Institute of Geochemistry, Chinese Academy of Science using a Thermo Fisher K-Alpha instrument fitted with Al Kα (hv = 1486.6 eV) at 300 W on, and at the pass energy of 100 eV for a better result in quantitative determination. In addition, a high vacuum condition of less than $3 \times 10^{-9}$ Pa, high resolution of 0.50 eV, and pass energy of 30 eV were set up for gaining precise binding energies in qualitative narrow scanning analysis. The calibration of the binding energy was referenced on the adventitious carbon (284.8 eV). The XPS spectrum was processed using the software Thermo Avantage V5.934.

## 3. Result

### 3.1. Petrography and Mineralogy of the Uraninite

The back-scattered electron (BSE) images were obtained by SEM (Figure 4). Figure 4a–c indicated that the uraninite grains generally showed a polygonal aspect with serrated edges, and mostly showed relatively homogeneous grayscale with a small number of broken cracks. Moreover, the crystal habits of the uraninite and mineralogical assemblages were identical to those previously described for primary uraninite at the MHK uranium deposit [13,39]. According to the BSE images, some coffinite was embedded in the uraninite, which showed different brightness in Figure 4d–g. The variations in BSE brightness indicated different chemical compositions [40]. The darker areas linked to lower average atomic weight reflected some changes in the chemical composition of the uraninite by a substitution of the heavy element (U) with a lighter element (Si). Furthermore, these in situ elemental substitutions in the uraninite were strong, but progressive, and probably ultimately led to a total conversion of uraninite to secondary uranium minerals such as coffinite [41,42]. The chemical composition of the uraninite would be described in more detail in the following sections.

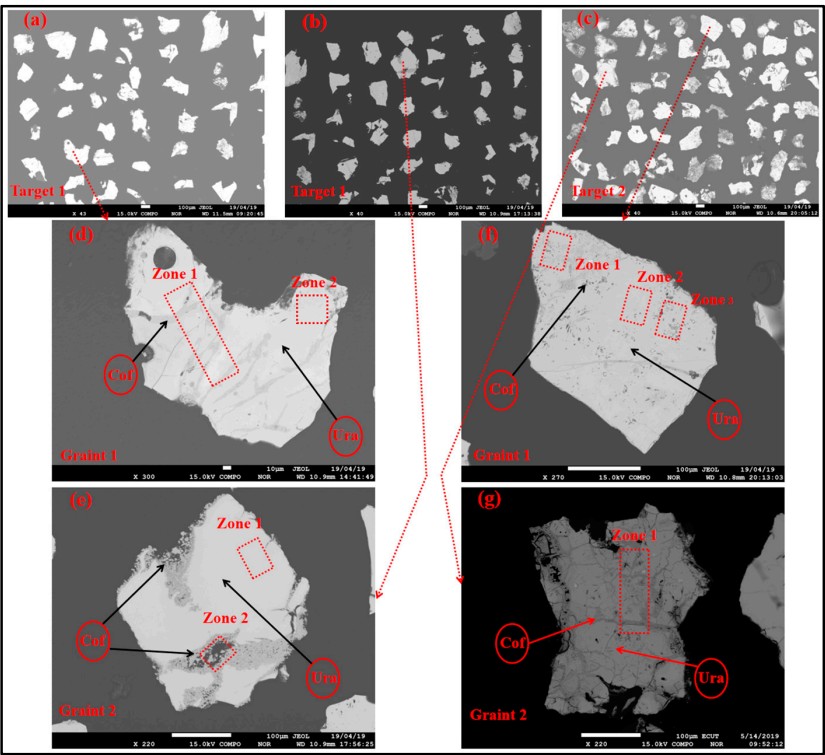

**Figure 4.** BSE (**a**–**g**) photomicrographs of typical uraninite from the MHK uranium deposit. (**a**,**b**) Viewshed map of Target 1. (**c**) Viewshed map of Target 2. (**d**,**e**) Typical grains of Target 1. (**f**,**g**) Typical grains of Target 2. "Ura" and "Cof" represent uraninite and coffinite, respectively. The analysis regions are shown in red rectangles for EDS mapping.

To analyze the major elemental composition of the uranium mineral, EDS was used as a semiquantitative tool to determine the percent of major components (the analytical zones were shown in Figure 4). Figure 5 lists the elemental constitution of each sample with their mass and atomic proportions. SEM-EDS analyses revealed that U, O, Si, and Ca were the major elements of the uranium mineral. However, Al, Fe, and especially Y, were only detected in some grains (Figure 5).

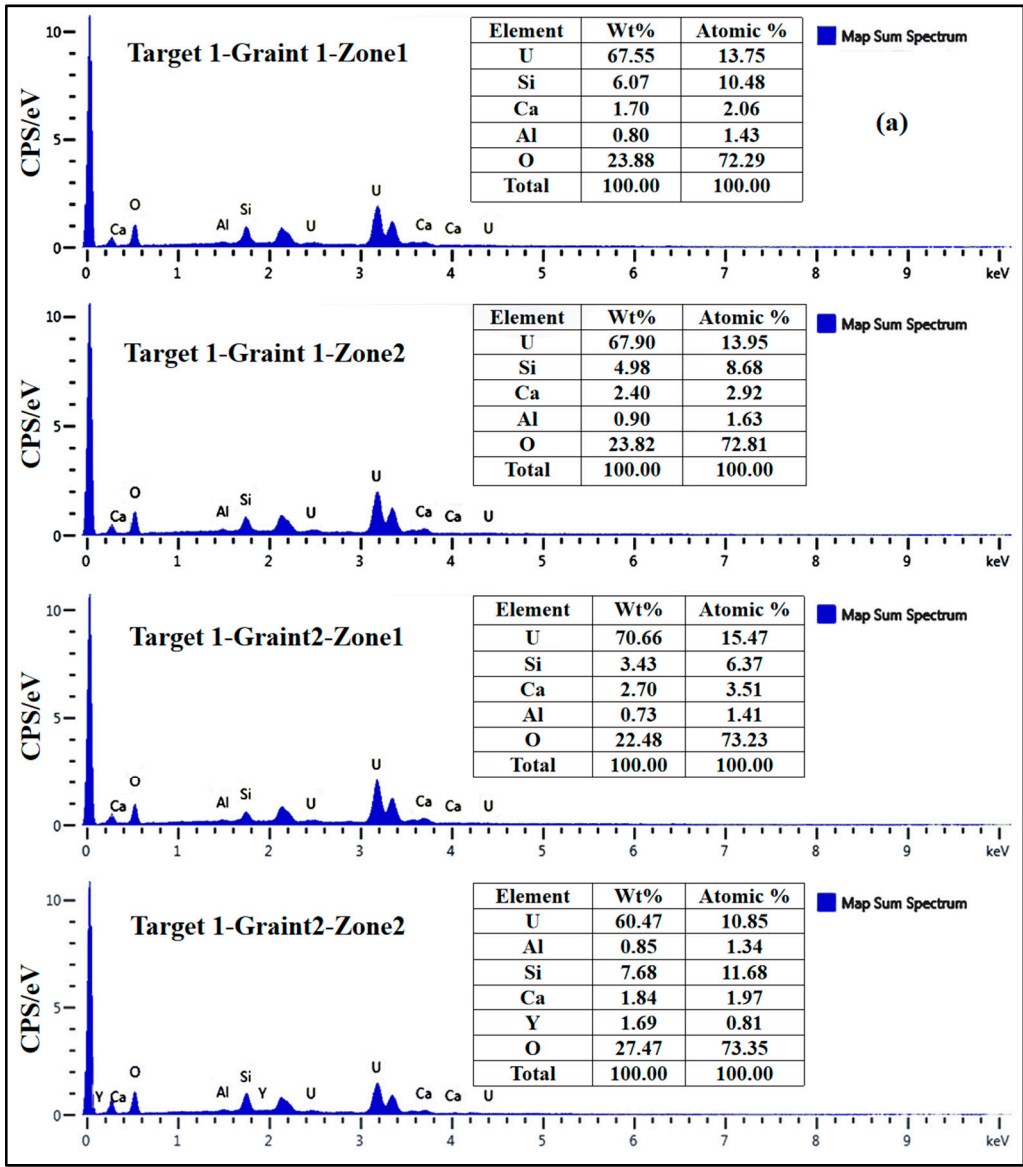

**Figure 5.** *Cont.*

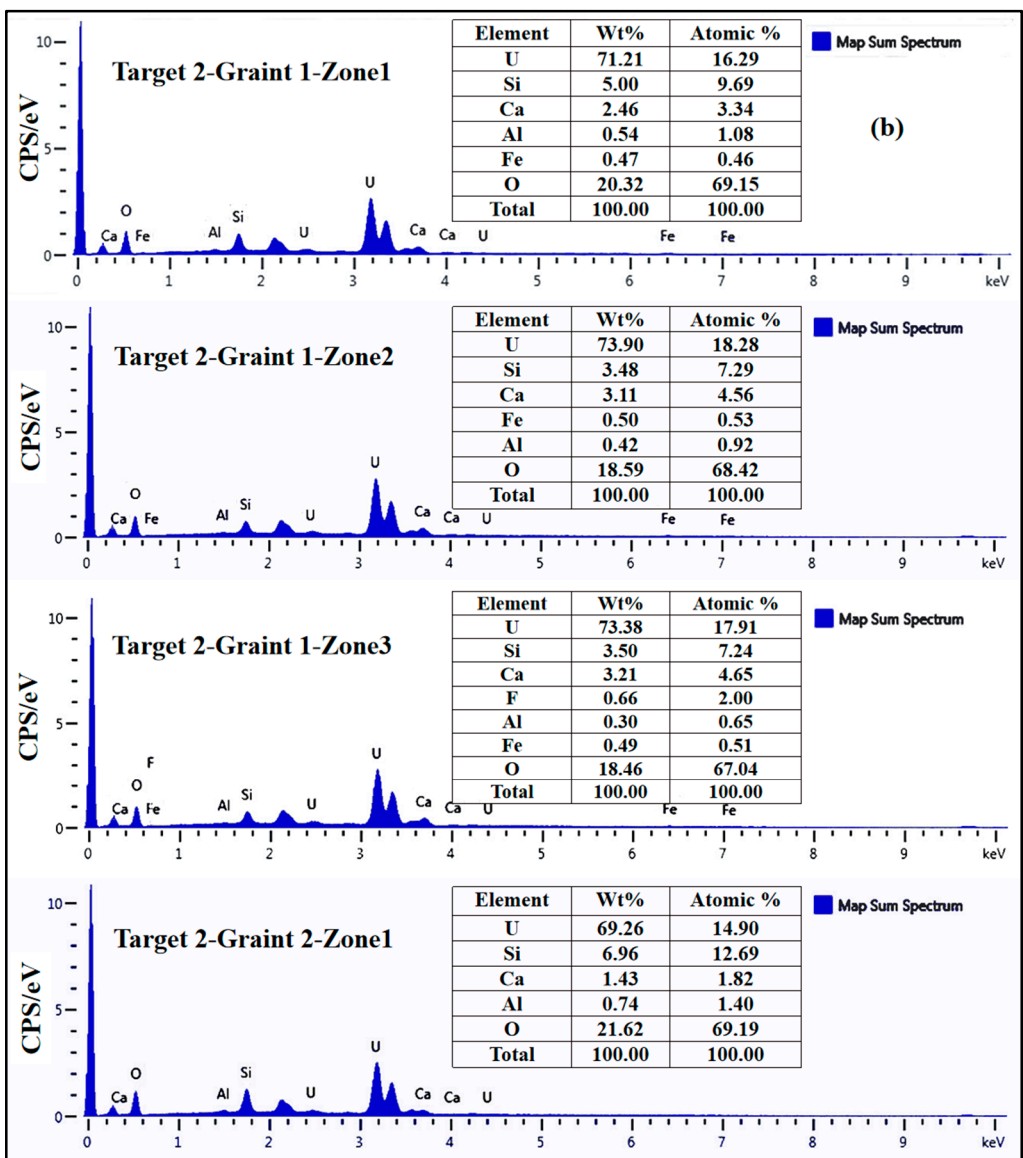

**Figure 5.** EDS mapping of the uraninite and related concentrations of elements from EDS. (**a**) The EDS mapping of Target 1. (**b**) The EDS mapping of Target 2. The peak value was the Au, which was coated to create a conductive surface.

### 3.2. Geochemical Composition of Uraninite

The quantitative elemental composition of uraninite were investigated by an electron probe microanalyzer (EPMA). The in situ chemical composition data are presented in Table S1. Moreover, the statistical characteristic value of the elemental composition of representative grains is shown in Table 1. The uraninite corresponding to the bright zones in the BSE-image (Figure 4) was characterized by the higher U, Ca, and Fe contents and lower Si and Al contents. Coffinite (the gray areas in BSE-image) showed relatively higher Si and Al contents. Concentrations of $UO_2$, $SiO_2$, $CaO$, $FeO$, and $Al_2O_3$ in uraninite and coffinite were 79.46 ± 2.03 and 65.79 ± 2.45 wt%, 6.19 ± 1.36 and 15.72 ± 1.70 wt%, 5.09 ± 0.80 and 2.08 ± 0.28 wt%, 0.55 ± 0.12 and 0.15 ± 0.07 wt%, and 0.57 ± 0.17 and 0.93 ± 0.20 wt%, respectively. In addition to the aforementioned oxides, the uraninite and coffinite also contained minor amounts of $Ce_2O_3$, $Na_2O$, $TiO_2$, $P_2O_5$, PbO, etc.

**Table 1.** The statistical characteristic value of oxide concentrations (wt%) of uranium mineral from the MHK uranium deposit (n.d = not detected; Spot no. "1-1" represents the number of target and grain, respectively).

| Spot No. | Uranium Mineral | Number of Measuring Points | Characteristic Value | $UO_2$ | $SiO_2$ | CaO | FeO | $Al_2O_3$ | $Ce_2O_3$ | $Na_2O$ | $TiO_2$ | $P_2O_5$ | PbO | MgO | $ZrO_2$ | NiO | $K_2O$ | $ThO_2$ | Total |
|---|---|---|---|---|---|---|---|---|---|---|---|---|---|---|---|---|---|---|---|
| 1-1, 1-2 | Uraninite | 25 | Maximum | 81.81 | 10.07 | 5.56 | 0.72 | 1.06 | 0.52 | 0.30 | 0.33 | 0.08 | 0.58 | 0.03 | 0.04 | 0.05 | n.d | n.d | 93.85 |
| | | | Minimum | 73.91 | 4.94 | 3.09 | 0.33 | 0.47 | 0.13 | 0.06 | 0.09 | 0.01 | 0.01 | n.d | n.d | n.d | n.d | n.d | 90.00 |
| | | | Average | 79.15 | 6.52 | 4.69 | 0.52 | 0.64 | 0.31 | 0.15 | 0.21 | 0.04 | 0.12 | 0.01 | 0.02 | 0.02 | n.d | n.d | 92.41 |
| | | | Standard Deviation | 2.51 | 1.60 | 0.76 | 0.11 | 0.18 | 0.10 | 0.05 | 0.09 | 0.02 | 0.16 | 0.01 | 0.02 | 0.02 | n.d | n.d | 1.30 |
| | Coffinite | 11 | Maximum | 70.94 | 18.42 | 2.64 | 0.30 | 1.24 | 0.60 | 0.21 | 0.14 | 0.53 | 0.61 | 0.09 | 0.03 | 0.03 | n.d | n.d | 87.95 |
| | | | Minimum | 61.52 | 12.49 | 1.72 | 0.06 | 0.69 | 0.13 | n.d | 0.03 | 0.07 | n.d | n.d | 0.02 | 0.01 | n.d | n.d | 83.71 |
| | | | Average | 65.71 | 15.64 | 2.08 | 0.15 | 0.98 | 0.35 | 0.07 | 0.06 | 0.27 | 0.13 | 0.03 | 0.03 | 0.02 | n.d | n.d | 85.38 |
| | | | Standard Deviation | 2.51 | 1.77 | 0.32 | 0.08 | 0.17 | 0.16 | 0.07 | 0.05 | 0.17 | 0.24 | 0.03 | 0.01 | 0.01 | n.d | n.d | 1.21 |
| 2-1 | Uraninite | 17 | Maximum | 81.18 | 8.02 | 6.16 | 0.77 | 0.55 | 0.48 | 0.23 | 0.15 | 0.06 | 0.09 | 0.04 | 0.05 | 0.09 | n.d | n.d | 93.92 |
| | | | Minimum | 77.35 | 4.95 | 4.49 | 0.36 | 0.39 | 0.17 | 0.07 | 0.07 | 0.01 | 0.01 | 0.01 | 0.01 | n.d | n.d | n.d | 91.43 |
| | | | Average | 79.93 | 5.70 | 5.67 | 0.59 | 0.46 | 0.30 | 0.14 | 0.10 | 0.03 | 0.05 | 0.02 | 0.03 | 0.04 | n.d | n.d | 93.01 |
| | | | Standard Deviation | 0.87 | 0.68 | 0.41 | 0.11 | 0.06 | 0.08 | 0.04 | 0.02 | 0.02 | 0.03 | 0.01 | 0.02 | 0.04 | n.d | n.d | 0.62 |
| | Coffinite | 3 | Maximum | 69.16 | 17.13 | 2.19 | 0.19 | 0.96 | 0.54 | n.d | n.d | 0.25 | n.d | 0.05 | n.d | n.d | n.d | n.d | 87.32 |
| | | | Minimum | 63.93 | 14.07 | 1.99 | 0.10 | 0.46 | 0.12 | n.d | n.d | 0.10 | n.d | 0.02 | n.d | n.d | n.d | n.d | 83.95 |
| | | | Average | 66.08 | 16.05 | 2.07 | 0.14 | 0.74 | 0.33 | n.d | n.d | 0.18 | n.d | 0.03 | n.d | n.d | n.d | n.d | 85.67 |
| | | | Standard Deviation | 2.74 | 1.71 | 0.11 | 0.05 | 0.25 | 0.21 | n.d | n.d | 0.08 | n.d | 0.02 | n.d | n.d | n.d | n.d | 1.68 |

High-resolution EPMA elemental mapping of the selected elements (major elements of U, Si, Ca, and Fe, and trace elements of Al, Ce, Na, and Pb) were also carried out to examine different grains and within-grain compositional heterogeneity. It showed that the four uranium mineral grains presented relatively variable element contents, especially exemplified by U, Si, Al, and Pb maps (Figure 6a–d and Figure S1a–d in Supplementary Materials). Additionally, internal heterogeneity within an individual grain was revealed by the element mapping, which was well corroborated with the stripe type distribution of U, Si, Ca, and Fe (Figure 6a–d). Importantly, it was vividly demonstrated that the ultimate conversion from uraninite to coffinite was characterized by a Si increase, coupled with a significant drop in the U, Ca, and Fe contents in these maps.

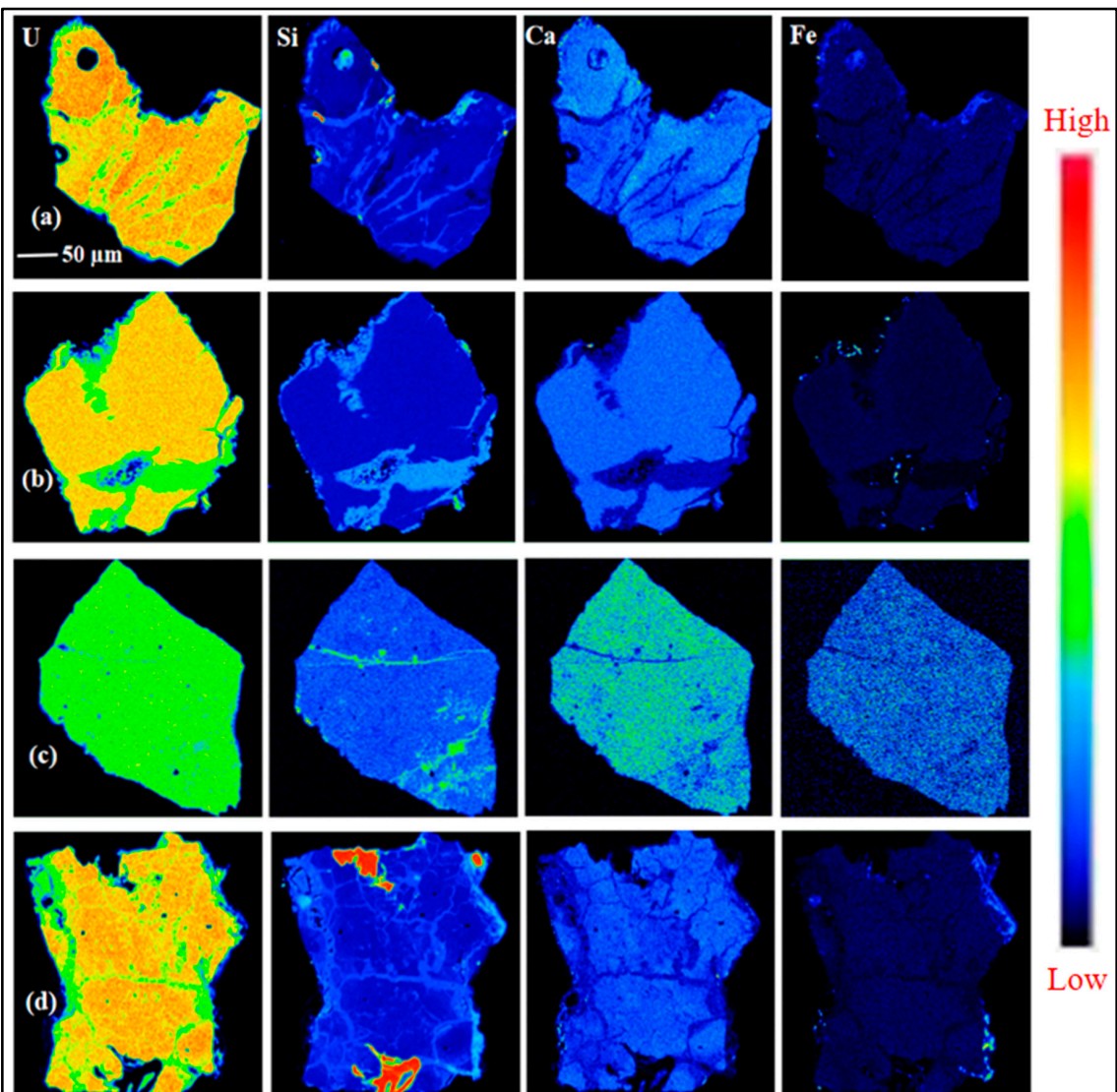

**Figure 6.** EPMA elemental mappings for U, Ca, Pb, and Si of uraninite grains. (**a**) Grain 1 of Target 1; (**b**) grain 2 of Target 1; (**c**) grain 1 of Target 2; and (**d**) grain 2 of Target 2 (warmer color indicates a higher concentration).

### 3.3. The Age of Uraninite from the MHK Uranium Deposit

In situ U–Th–Pb chemical ages of fine uranium mineral can be obtained through EPMA due to its high spatial resolution [15,16]. Single-point U–Th–Pb chemical ages of uranium mineral grains were calculated through the Cameron–Schiman's Equation (Equation (1)), which was described in detail

by Bowles [43] and applied by uranium deposit geologists [40,44,45]. Moreover, the accuracy of Pb analyses by EPMA was about 0.01%, which corresponded to errors of 1 Ma for the chemical ages.

$$t(y) = Pb \cdot 10^{10}/(1.612U + 4.95Th) \tag{1}$$

In this equation, amounts of U, Th, and Pb were contents in atomic percentage (%) and the resultant age was in years. Through the calculation, the average chemical ages of uraninite and coffinite were 5.96 ± 3.49 Ma and 5.32 ± 2.92 Ma, respectively, except for several extreme values (Figure 7).

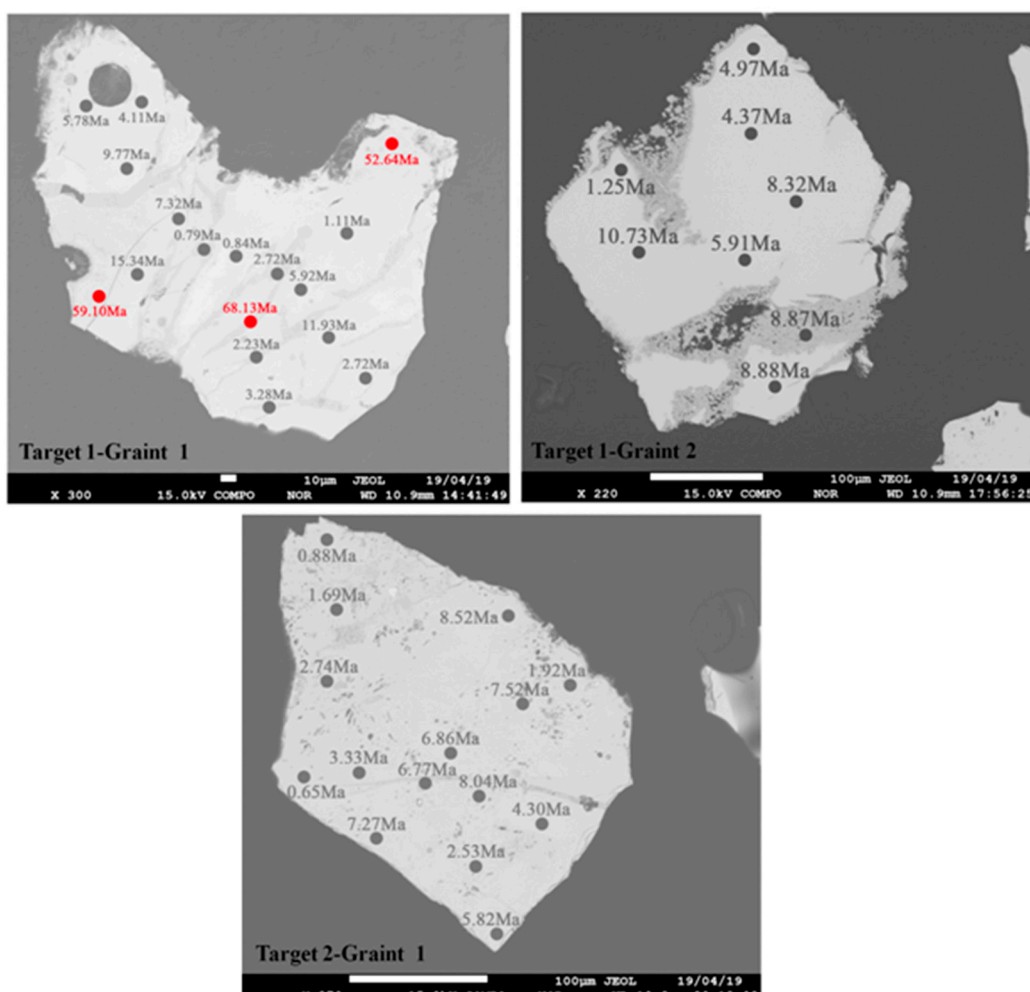

**Figure 7.** Calculated chemical ages of uraninite and coffinite from the MHK uranium deposit. The red circles were three extremely high values.

*3.4. Acid Hydrometallurgy of Uraninite*

The distribution of XPS measured spots are shown in Figure S2, and grains 1–3 were analyzed again when the uraninite samples were leached by sulfuric acid. In general, the abscissa and ordinate of the XPS curve were binding energy and measured intensity, respectively. The handbook or database for XPS binding energy could be applied to identify elements qualitatively [46]. As shown in the XPS spectrum, the binding energy at 284.88 eV corresponded to the C 1s peak, and O 1s peak, U 4f peak, Si 2p peak, and Ca 2p peak were at 533.73 eV, 381.07 eV, 97.16 eV, and 347.37 eV, respectively. Thus, the whole spectrum scanning indicated that the uraninite mainly consisted of C, O, U, Si, and Ca with minor Ce (Figure 8). These results were well consistent with the determination by EPMA and EDS.

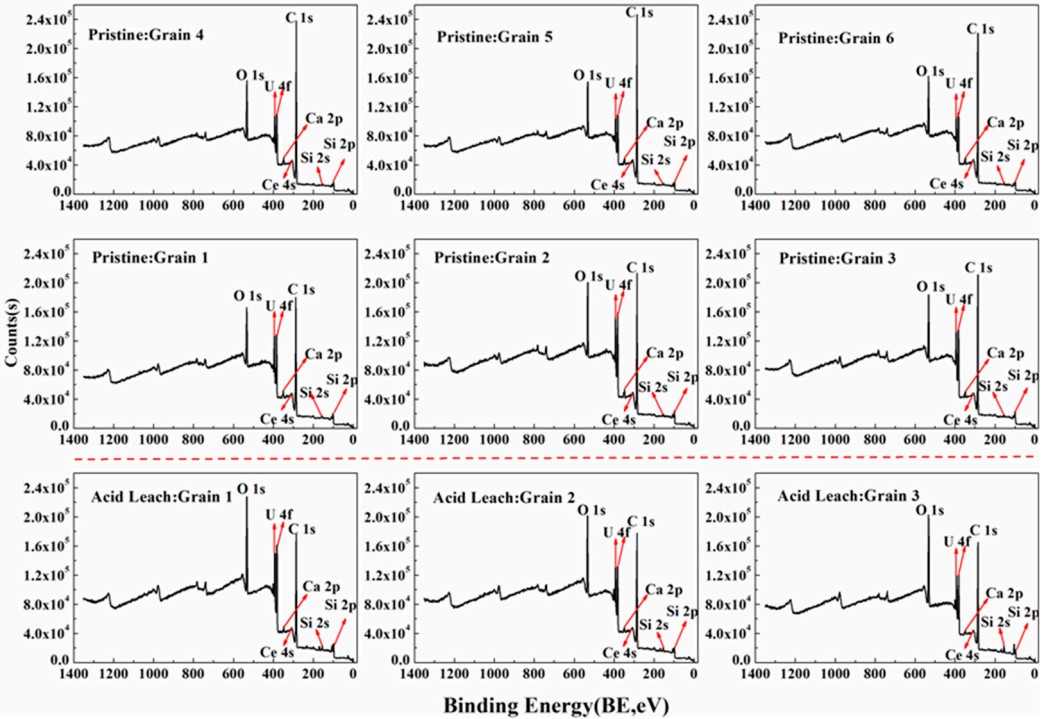

**Figure 8.** Survey XPS scan of the uraninite in the MHK uranium deposit. Above the red dotted line: pristine samples; below the red dotted line: samples leached by sulfuric acid). The red arrows pointed to the names of elements peak.

In order to identify the major ionic species of uranium of uraninite, the XPS narrow scan of U 4f (the spin orbit splitting was 10.8 eV) was done repeatedly. Moreover, the U $4f_{7/2}$ bands were further resolved to quantitatively assess the existing uranium species. The peak fitting curves of the different uranium chemical valences are shown in Figure 9. These results indicate that uranium in the uraninite existed in the forms of $UO_2$, $U_3O_8$, and $UO_3$ with a binding energy of 380.0 eV, 381.0 eV, and 381.9 eV, respectively [39,47].

The percentages of uranium species of different uraninite grains are listed in Table 2. It indicated that the occurrence of uranium chemical valences on the surface of different particles were different, despite the similarities in the uranium species before and after the acid leaching. In these samples, $U_3O_8$ and $UO_3$ are the main forms of uranium, which suggest that uranium existed mostly at high valence states. Furthermore, oxygen coefficients ($k_o$) of all grains were also calculated according to their percentages of uranium in various states (Table 2).

**Table 2.** Summary of oxygen coefficient $k_O$ and $UO_{2+x}$ oxide mass percentage of uraninite (%; n.d = not detected).

| Spot No. | Oxygen Coefficient, $k_0$ | Mass Percentage, % | | |
|---|---|---|---|---|
| | | $UO_2$ | $U_3O_8$ | $UO_3$ |
| Grain 1 (pristine) | 2.72 | n.d | 83.65 | 16.35 |
| Grain 1 (acid leaching) | 2.79 | n.d | 60.95 | 39.05 |
| Grain 2 (pristine) | 2.84 | 15.94 | n.d | 84.06 |
| Grain 2 (acid leaching) | 2.91 | 9.30 | n.d | 90.70 |
| Grain 3 (pristine) | 2.90 | 9.92 | n.d | 90.08 |
| Grain 3 (acid leaching) | 2.93 | 6.99 | n.d | 93.01 |
| Grain 4 (pristine) | 2.89 | 10.89 | n.d | 89.11 |
| Grain 5 (pristine) | 2.78 | n.d | 65.32 | 34.68 |
| Grain 6 (pristine) | 2.85 | 15.15 | n.d | 84.85 |

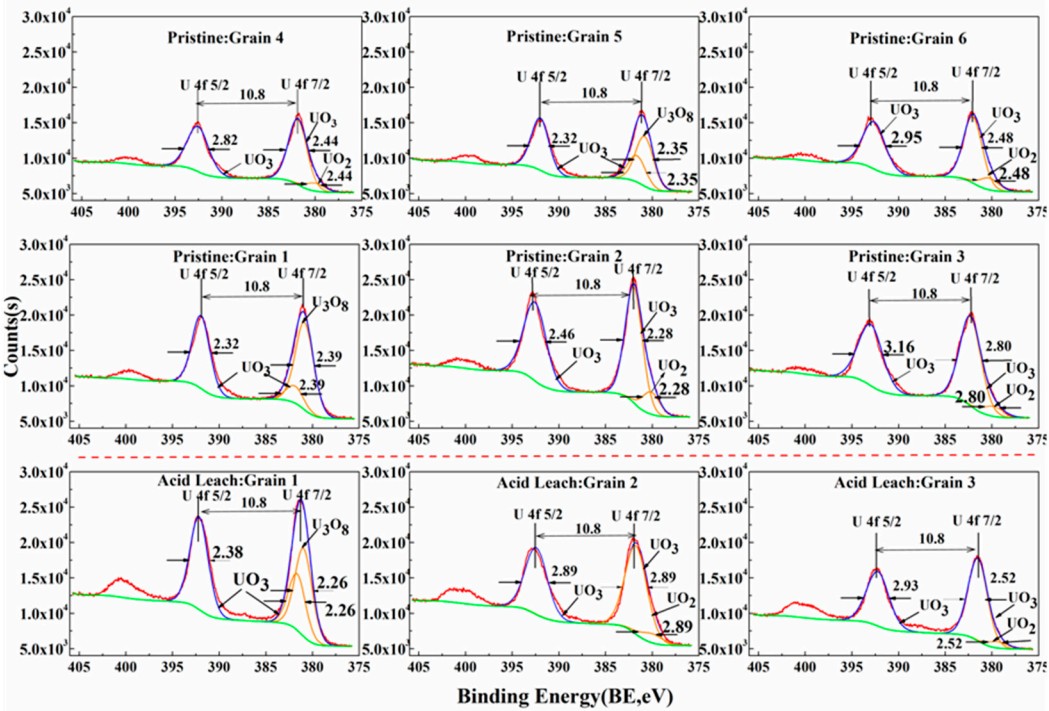

**Figure 9.** XPS narrow scans of uranium in the MHK uranium deposit. Above the red dotted line: pristine samples; below the red dotted line: samples leached by sulfuric acid. The red lines, blue lines, orange lines and green lines were original spectrogram, fitting spectrogram, peak fitting curves of different uranium chemical valences and baseline, respectively.

## 4. Discussion

### 4.1. Chemical Composition of Uraninite

As shown in Table 1, the analytical data fell short of 100%. This could be interpreted by the presence of structural water, undetected rare earth elements (REEs) as well as the hexavalent uranium due to the U completely calculated as $UO_2$ [16,41,48,49]. According to previous studies, low-temperature hydrothermal uraninite was marked by U/Th > 1000 [50]. In this study, the contents of $ThO_2$ were extremely low (mostly below the limit of detection and U/Th were far greater than 1000), which were in accordance with their hydrothermal origin due to the generally lower solubility of Th at low to intermediate fluid temperature (i.e., lower than 300 °C as previously mentioned) [15,17,51]. However, the uraninite and coffinite contained relatively high calcium contents (up to 6.16 wt% CaO), which were also in agreement with their hydrothermal origin [16]. In fact, it is commonly considered that the incorporation of Ca into the uraninite structure compensates for the charge difference because of the presence of hexavalent uranium [52]. Therefore, the concentration of Ca correlates positively to that of U in the mineral grains (Figure 10a). Furthermore, calcium was most possibly incorporated during crystallization, even if a small part of it could be related to post-crystallization alteration [13]. Both Si versus U and Si versus Ca of uraninite and coffinite showed significantly negative correlations (Figure 10b,c). As shown in the BSE images (Figure 4), uraninite was possible intergrown with coffinite. And then, the most significant difference about uraninite and coffinite was up to 18.42 wt% $SiO_2$, and low to 63.93 wt% $UO_2$ and 1.72 wt% CaO, respectively. The obvious trends of high U and Ca and low Si from uraninite to coffinite are shown in Figure 11. As a matter of fact, these characteristics of the two minerals' composition generally depended on the features of the host rocks, and amounts and compositions of the alteration fluids, particularly their capacity for oxidation [41].

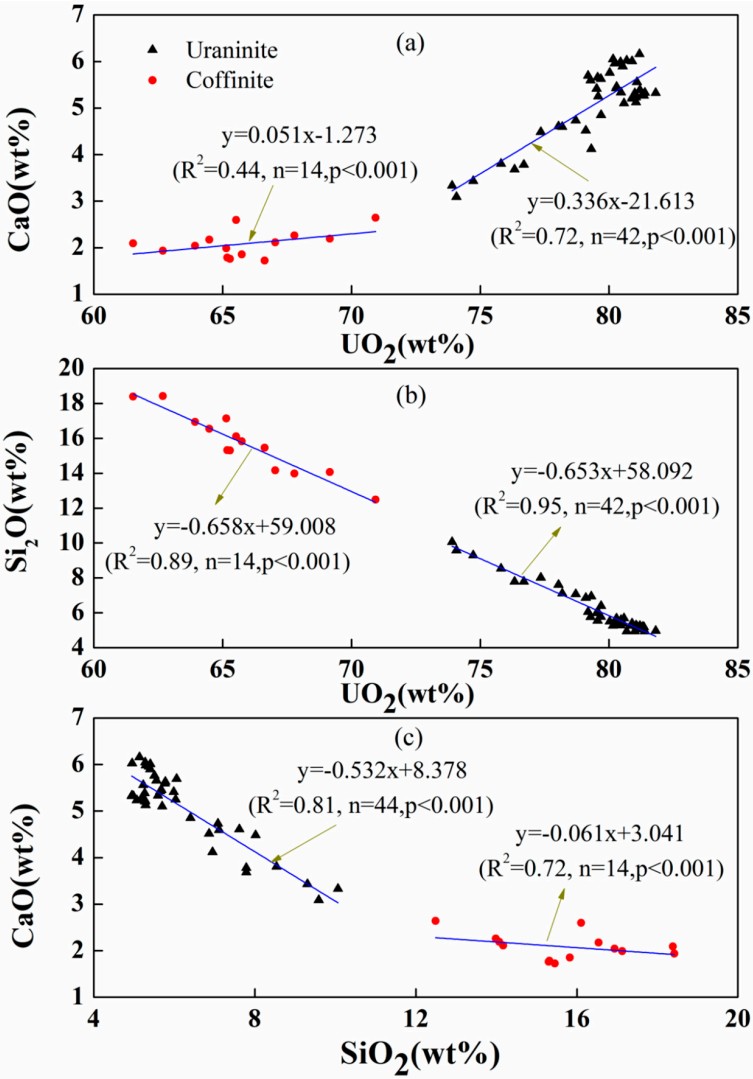

**Figure 10.** (**a**) CaO vs. $UO_2$, (**b**) $SiO_2$ vs. $UO_2$, and (**c**) CaO vs. $SiO_2$ plots of uraninite and coffinite.

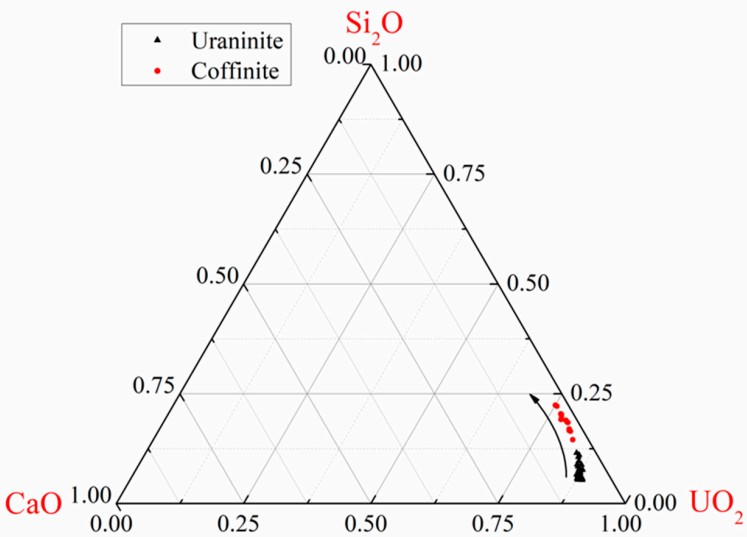

**Figure 11.** The significant trends indicated an overall U, Ca loss, and Si gain during metasomatism.

### 4.2. Pb Loss and Reliability of the Uraninite U–Th–Pb Chemical Ages

Chemical U–Th–Pb ages derived from the composition of uraninite varied from 0.65 to 68.13 Ma, indicating Pb loss [53]. In this study, Pb rich zones were clearly seen predominately along the grain edge (Figure S1), which meant that initial Pb existed. Moreover, there was a low correlation of $UO_2$ vs. PbO (Figure 12) and extremely weak correlations of $\Sigma(SiO_2 + CaO + FeO)$ vs. chemical ages of uraninite, which were also in agreement with obvious Pb loss in uraninite [15,40,54]. A well-defined negative correlation between the $\Sigma(SiO_2 + CaO + FeO)$ contents and $\Sigma(UO_2 + PbO)$ contents (Figure 13) indicated that the major elements ($UO_2$ + PbO) had been partially replaced by elements of $\Sigma(SiO_2 + CaO + FeO)$ [41]. As we all know, uraninite usually has a high content of U, so a large number of $\alpha$ particle would be released during the decay, leading to lattice damage. Next, Ca, Si, Fe, and other impurity elements would enter into the crystal lattice of uraninite because of the effect of the alteration fluids later, which led to U and Pb loss [55]. Furthermore, Pb loss, U redistribution, and incorporation of Si, Ca, and Fe in uraninite suggested that it was an opening of the U–Th–Pb system [48]. However, the mechanism of Pb loss is unclear and Pb loss quantification is lacking in this area. It may have been continuous diffusional Pb loss, episodic recrystallization, or by the formation of secondary uraninite [52,56].

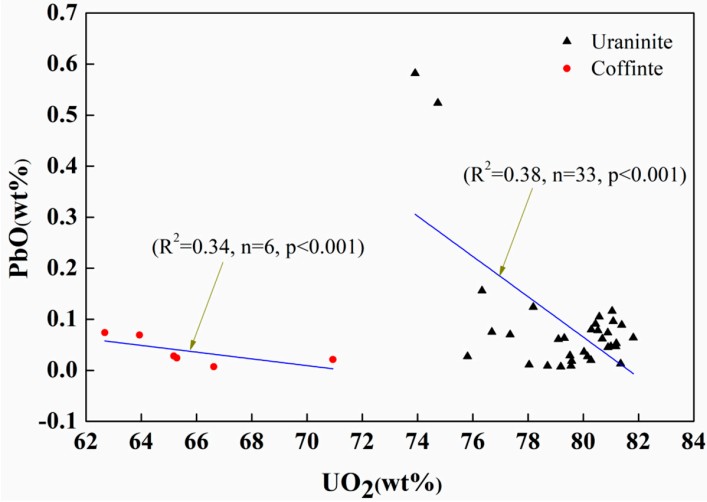

**Figure 12.** PbO–$UO_2$ correlation trend of uraninite and coffinite grains.

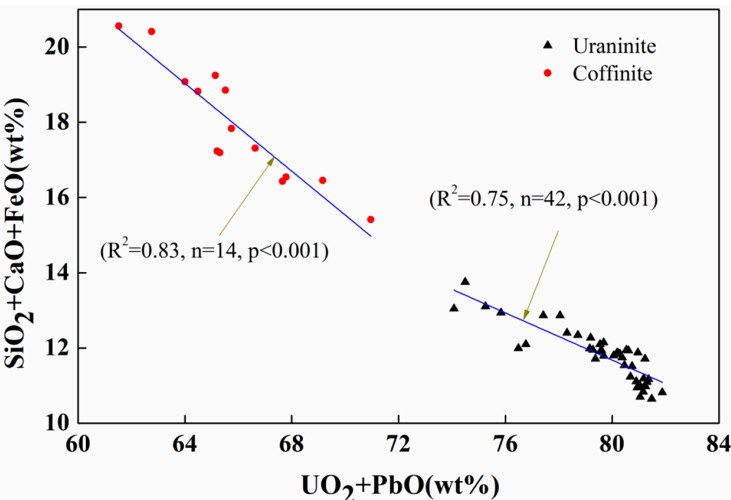

**Figure 13.** Binary diagrams of $UO_2$ + PbO vs. $SiO_2$ + CaO + FeO.

Due to the Pb loss of uraninite, a "chemical age" of uraninite calculated from the EPMA analyses were highly questionable [57–59]. Therefore, uraninite in situ Laser Ablation-Inductively Coupled Plasma-Mass Spectrometry (LA-ICP-MS) U–Pb geochronological research should be carried out, in order to obtain a precise mineralization age for the MHK (No. 302) uranium deposit. Moreover, it is also worthwhile to research the Pb loss mechanism for the crucial uranium mineralization age, especially for deciphering the mechanism of multi-stage uranium mineralization in South China.

### 4.3. Acid Leaching Accessibility of Uraninite

Until now, leaching by utilizing sulfuric acid solutions is still the most widely used process for extracting uranium from uraninite, coffinite, and brannerite ores due to the relatively low cost and wide availability of the acid [5]. In general, the acid sulfate leaching process includes treating the uranium mineral with sulfuric acid at elevated temperatures (usually 30–50 °C), with an oxidant to convert the low-valent state uranium to the hexavalent state uranium [60]. Moreover, chemistry of uranium ores such as element species will change during acid sulfate leaching, and it has been discussed in detail by some authors [61,62]. In this study, the variation of element composition of uraninite before and after acid leaching is shown in Figure 14. It suggests that the mass percentages of U, C, and Ca decreased significantly, followed by O increasing observably and S appeared when the uraninite target was leached by sulfuric acid. However, the content of Si did not show obvious variation characteristics.

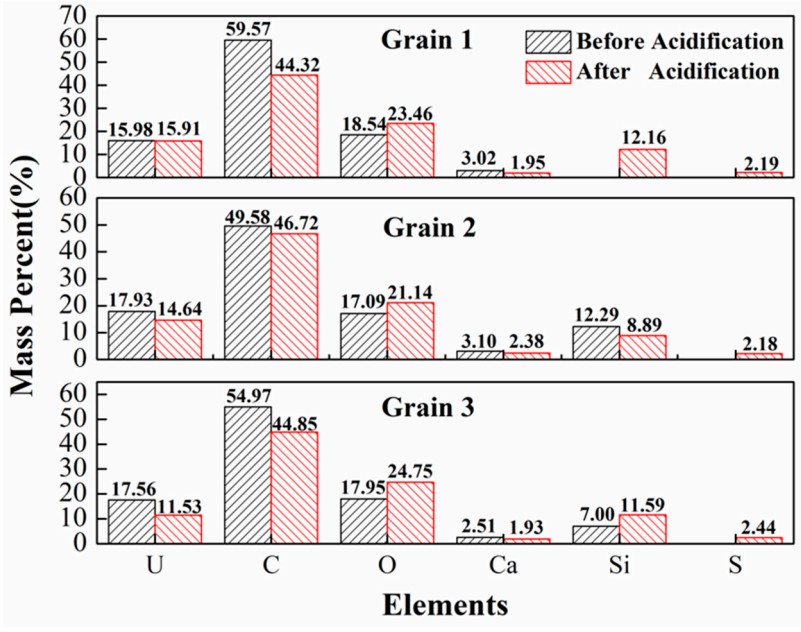

**Figure 14.** Variation characteristics of elements before and after acid leaching.

All the uranium mass percentages of different sized grains (grains 1, 2 and 3) showed a decrease in the low-valent state ($UO_2$ and $U_3O_8$) and a corresponding increase in the high-valent state ($UO_3$) in the proportion when these samples were leached by sulfuric acid (Figure 15). This indicates that uranium on the surface of uraninite was partially oxidized during the acid leaching process. Consequently, the most likely reaction pathway was the oxidation of tetravalent uranium to hexavalent uranium because of the leaching [63]. Some studies indicated that the oxidation of $UO_2$ possibly proceeded through two successive electrochemical reactions, coupled with a $U^{5+}$ intermediate at the $UO_2$ surface noticeable between the $U^{4+}$ and $U^{6+}$ oxidation states [64,65], but it was not detected in this study. In addition, the binding energy of the uranium oxide tended to rise due to the increase in the oxygen coefficient ($k_o$). Hence, the main peak corresponding to U $4f_{7/2}$ moved 1.28 eV to the left in the XPS spectrum (Figure 16). It was worth noting that uraninite in this deposit hosted predominantly oxidized

uranium, which was novel and unusual because it contained uranium as a necessary structural constituent in either the reduced $U^{4+}$ or oxidized $U^{6+}$ valence state.

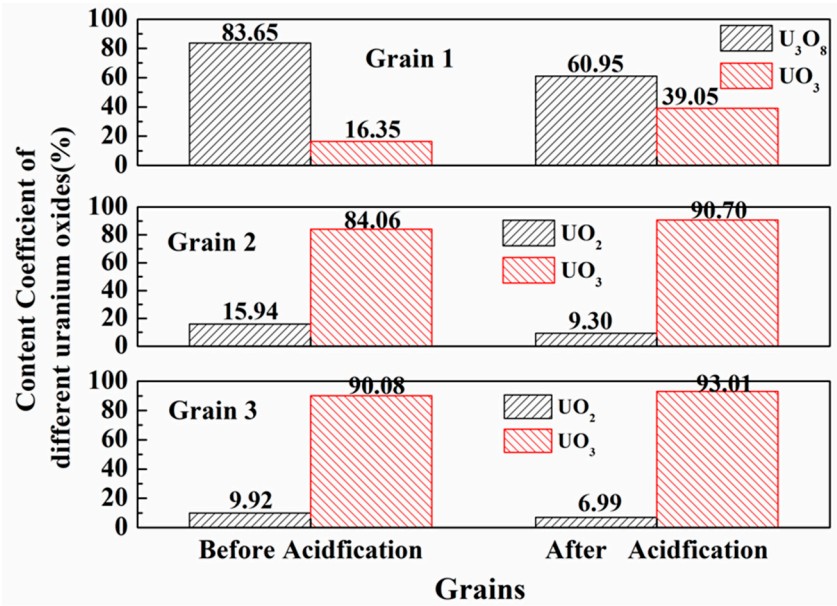

**Figure 15.** The variation of the uranium valence after sulfuric acid leaching.

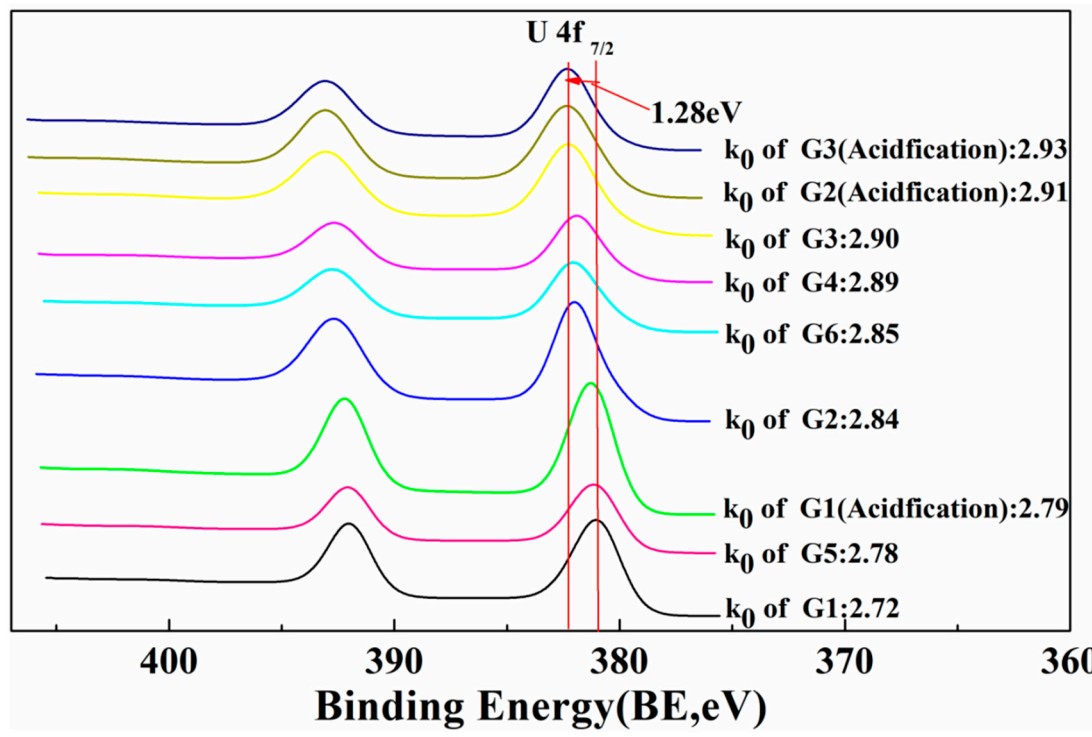

**Figure 16.** The variation of the U 4f spectra along with oxygen coefficient ($k_0$).

In light of the above analysis, it is possible to extract uranium from uraninite in the MHK deposit by using an acid sulfate leaching process. In order to raise the uranium extraction rates, it is necessary and important to appropriately improve sulfuric acid concentration typically used in commercial leaching operations [66]. An understanding of the low temperature aqueous and redox geochemistry of uraninite provides important background information as to the stability of uraninite under different environmental conditions (e.g., T, pH, and $pO_2$), all of which are relevant to understanding the behavior

of uraninite in commercial leaching processes. However, the finding that uraninite at this deposit hosts predominantly oxidized uranium is novel and unusual, and must be discussed in great detail and from a mineralogical and crystallographic point of view.

## 5. Conclusions

The present study provided the uranium mineralogy, chemical compositions, U–Th–Pb chemical age, and uranium valence of uraninite in the Mianhuakeng granite-hosted uranium deposit, South China. In addition, the variation characterization of the compositions and uranium valence after being acid leached were particularly discussed in detail. The following conclusions were drawn with regard to the surface chemical properties and acid leaching accessibility of uraninite in the MHK deposit.

The uraninite was intergrowth with coffinite. Elemental contents of the two minerals were significantly different. Their major elements concentrations for $UO_2$, $SiO_2$, and CaO were 79.46 ± 2.03 and 65.79 ± 2.45 wt%, 6.19 ± 1.36 and 15.72 ± 1.70 wt%, and 5.09 ± 0.80 and 2.08 ± 0.28 wt%, respectively. Then, the most important difference between uraninite and coffinite was up to 18.42 wt% $SiO_2$, and as low as 63.93 wt% $UO_2$ and 1.72 wt% CaO, respectively. The average chemical ages of uraninite and coffinite were 5.96 ± 3.49 Ma and 5.32 ± 2.92 Ma, respectively. However, the "chemical age" calculated from the EPMA analyses were highly questionable due to lead loss.

Second, uranium in the uraninite generally existed in the forms of $UO_2$, $U_3O_8$, and $UO_3$ with a binding energy of 380.0 eV, 381.0 eV, and 381.9 eV, respectively. The fractions of $U_3O_8$ and $UO_3$ were overwhelming, which suggested that uranium mostly existed at high valence states. Moreover, uranium on the surface of uraninite was partially oxidized by sulfuric acid leaching, which led to a conversion of tetravalent uranium to hexavalent uranium. Thus, uraninite in the MHK deposit was able to be leached by sulfuric acid.

**Supplementary Materials:** The following are available online at http://www.mdpi.com/2075-163X/10/9/747/s1, Table S1: Representative analyses of the chemical composition (wt%) and single-spot ages of uranium mineral from the MHK uranium deposit (Ma); Figure S1: EPMA elemental maps for Al, Ce, Na, and Pb of uraninite grains; Figure S2: Locations of the spot analyses for the XPS.

**Author Contributions:** Conceptualization, Z.S.; Methodology, Z.S. and G.L.; Project administration, Z.S.; Funding acquisition, Z.S.; Formal analysis, J.W. (Jian Wang); Investigation and sampling, K.Z., J.W. (Jiaxin Wei), L.X. and J.W. (Jian Wang); Data curation, J.W. (Jian Wang); Writing-original draft preparation, J.W. (Jian Wang); Writing-review and editing, G.L., Y.L., X.W., Z.Z. (Zhongkui Zhou), Z.Z. (Zhihong Zheng), and Y.Z. All authors have read and agreed to the published version of the manuscript.

**Funding:** This research was supported by the National Natural Science Foundation of China (No. 41772266, 51764001) and the Open Fund from the State Key Laboratory of Nuclear Resources and Environment, East China University of Technology (No. NRE1929).

**Acknowledgments:** This research was carried out at the State Key Laboratory of Nuclear Resources and Environment, East China University of Technology. Specially, we thank the laboratory members for helping to analyze the mineral samples by EPMA and SEM-EDS. We are very grateful for the Guangzhou Institute of Geochemistry, Chinese Academy of Science for helping with the surface analysis of the mineral samples through XPS.

**Conflicts of Interest:** The authors declare no conflict of interest.

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
