# Peer review of "Geochemistry and Acid Hydrometallurgy Accessibility of Uraninite from Mianhuakeng Granite-Hosted Uranium Deposit, South China"

_minerals, doi:10.3390/min10090747_

Round 1

Reviewer 1 Report

Overall the manuscript is clearly written (with the exception of the odd typo) and organized; it reads well. It contributed new data (uraninite chemistry and oxidation state of uranium) of limited usefulness and provides limited new understanding of uranium mineralogy. As it is, we know little new about these deposits or about uranium mineralogy that was not already known.

There are several shortcomings that should be addressed before this manuscript becomes a valuable contribution. These include:

The local geology and the deposit geology are not described at all. As it is, we know nothing of these deposits, which hampers interpretation. Figure 1 is not sufficient in itself to provide adequate geological context. A thorough and detailed description of the deposit and a corresponding figure must be added.

Never use the term "pitchblende": this is not an acceptable mineralogical name. In informal usage, it is often understood to mean "fine-grained, soothy uraninite", but in strict mineralogical sense, the mineral name that must be used is uraninite.

The images in Figure 2 d-g must be larger to be of any use. The same applies to Figures 3 and 5: the spectra are too small to be visible and understood. Same with Figure 4: the images are too small to be useful. The plots in Figure 10 are also too small to be useful. All of the above must be at least doubled in size.

The age calculations (lines 174-176) must be described in detail, as they are important. Just giving the references is insufficient.

The chemical ages are obviously unreliable: the overall variation (0.65 to 68.13 Ma) and the standard deviations (± 59 and ± 55%) are much to large to indicate that they can be of any use, contrary to what the authors suggest (line 274). Given the lack of a sustained relationship between chemical composition and chemical age (Figure 11), this is one of the cases where chemical ages cannot be used because they are too unreliable, and likely affected by strong Pb* loss (as also suggested by the low totals). The chemical ages obtained are lower than previous age estimates for these deposits (line 92, lines 180-181). Isotope ages (e.g., laser ablation ICP-MS U-Pb) should have been used instead. I suggest that the whole section on ages be removed, or at least heavily amended to state that the chemical ages are likely wrong.

The whole section on elemental maps (239-251, Figure 9) must be moved to Results: they have nothing to do with the Discussion.

The references are not up to date; they include too many local sources and neglect international literature. Some very important and relevant recent papers are not mentioned (e.g., Alexandre et al 2015, Frimmel et al 2014). The information and ideas in the most recent international literature must be incorporated in the interpretation of the data in this manuscript.

Reviewer 2 Report

Wang et. at., discuss the major element abundances in coffinite and pitchblende separates from the Mianhuakeng granite-hosted uranium deposit. The data looks robust and of good quality and the interpretations are sound. I have only a few minor comments below. However, I feel the English needs to be improved before the manuscript can be accepted (especially the introduction).

Figure 3. Using yellow for the spectrums makes it difficult to see both on the screen and when printed in black and white. 

Line 164: Mention the figure you are referring to i.e., figure 2.

Line 167: Suggest changing the word chemicals to oxides

Line 180: Those 3 spots (1-1-14, 1-1-17, 1-1-18) with ages > 50 Ma also have the highest Pb content.  From Figure S1 you can see the enrichment of Pb along the edges of the grain, which corresponds with 1-1-14 and 1-1-18. Suggesting Pb gain? Also, I think you should include what ages have been determined in the area, including the technique used to calculate those ages. 

Figure 9. The red is extremely difficult to see. It took me some time to notice it.

Line 258: You describe in great detail the metasomatism that has affected the samples in Fig 8 but here you state that it is a closed system with respect to U-Pb(-Th). In addition, with Figure S1 you are clearly seeing Pb rich zones predominately along the edge. How do you explain these relatively rich Pb zones if you are assuming no initial Pb? 

Figure 11. The figure description does not match the figure axes.

Reviewer 3 Report

Very nice study of the mineralogy of pitchblende and coffinite that I recommend for publication. There are a few problems, discussed below. The manuscript would greatly benefit by proofreading by a native English speaker. Some language is incorrect or confusing. In place analysis of pitchblende, in addition to analysis of mineral separates, would have been very beneficial to this study. Please expand the discussion of high Ca-content in relation to fluid source (line 223). The discussion of U and Ca loss and Si gain during metasomatism is unclear, how are the authors measuring the degree of metasomatism? How do the authors deduce that pitchblende is converting to coffinite. This texture seems that it could just as easily be described as intergrowths between the two minerals. Lines 268-9, please expand the description to indicate how the correlation of the elements shown shows a closed vs. open system with respect to U and Pb. I do not clearly see any cause and effect based on the manuscript as written. Figure 9 - the high degree of lead seen in element mapping would argue against this being as young a mineral as the authors suggest. Figure 12, unclear how the authors conclude that UO2 and U3O8 are being leached as opposed to UO3? As such, the derivative discussion in the rest of this paragraph is unfounded. Figure 13, title "Content Coefficient of Uranium" makes no sense, possibly your English-language proofreader can correct this. Conclusions - I am not convinced by the paper that pitchblende is being altered to coffinite, this argument needs to be made more thoroughly with support. I am not convinced the authors have proved that uranium on the surface of the grains is hexavalant uranium. The conclusion that pitchblende can be leached by sulfuric acid is not supported. Very little U was actually leached (See Figure 12). Would more U be available in an industrial application? If so show that data. 

Round 2

Reviewer 1 Report

The authors have addressed most of my initial comments and the manuscript is improved. However, there are still several significant schortcomings that must be addressed before this manuscript is ready for publication. The major of these changes are outlined below.

The text must be reviewed by a native English-speaking geologist. The proper geological terms much be used. FOr instance: line 92 and subsequent: Several of the terms are not appropriate: there is no such thing as "silk", or micaization, or feldsparization, or petrification. Line 102: lentil is food, not a geological term.

Deposit geology: It should be further extended. Information about the number of ore bodies, their size, the grade and tonnage must be included. A figure showing the deposit geology must be added.

Table 1: The chemical analyses must be reported in two digits (e.g., 12.34) past the decimal point, not three (e.g., 12.345) : the EMPA are not precise enough for three.

Chemical ages: A lot of space is devoted to the chemical ages, both in terms of results and discussion. However, the ages obtained are meaningless, as they are clearly affected by strong recent radiogenic lead loss. Therefore, all relevant parts of the manuscript should be significantly reduced (to something like 10% of the current length), or completely removed: they do not contribute any new knowledge or understanding about this deposit or this type of deposits, or of uraninite. For instance, section 4.2 can be reduced to one sentence: the chemical ages are meaningless.

The characteristics of the regression lines in Figures 10 and 11 (correlation coefficient, number of points, p?) are wrong and meaningless. Basically, all they demonstrate is the lack of correlations, in particular between chemical ages and chemical composition. It would have been much better to just add a correlation table of all elements vs. all elements including the chemical ages.

Oxidation state: This is the only part of this manuscript that is important and relevant and contributes something new to our understanding of uraninite. However, the authors devote only one short paragraph to this topic in the Discussion (lines 344-351), and even then, only from metalurgical point of view. However, the finding that uraninite at this deposit hosts predominantly oxidized uranium is novel and unusual and must be discussed in great detail and from a mineralogical and crystallographic point of view.

In conclusion, the manuscript should be rewritten, with most of the discussion about chemical ages removed and an extensive discussion about the oxydation state of uranium in uraninite added. Only then will this work represent a valuable and useful contribution to our understanding of uraninite.

Author Response

Dear Riewer,
